# Clinical Value of Novel Echocardiographic Biomarkers Assessing Myocardial Work in Acute Heart Failure—Rationale and Design of the “Beyond Myo-HF Study”

**DOI:** 10.3390/diagnostics13061191

**Published:** 2023-03-21

**Authors:** Vasileios Anastasiou, Stylianos Daios, Dimitrios V. Moysidis, Maria-Anna Bazmpani, Thomas Zegkos, Theodoros Karamitsos, Kali Makedou, Christos Savopoulos, Georgios Efthimiadis, Antonios Ziakas, Vasileios Kamperidis

**Affiliations:** 1First Department of Cardiology, AHEPA Hospital, School of Medicine, Aristotle University of Thessaloniki, 54636 Thessaloniki, Greece; 2Laboratory of Biochemistry, AHEPA Hospital, Faculty of Health Sciences, School of Medicine, Aristotle University of Thessaloniki, 54636 Thessaloniki, Greece; 3First Propedeutic Department of Internal Medicine, AHEPA Hospital, School of Medicine, Aristotle University of Thessaloniki, 54636 Thessaloniki, Greece

**Keywords:** acute heart failure, echocardiography, myocardial work, novel imaging biomarkers, response to treatment, prognosis

## Abstract

Background. Despite ongoing treatment advancements in chronic heart failure (HF), mortality and readmission rates remain high for patients hospitalized for decompensated acute HF. These patients represent a distinct HF group, which requires emergent echocardiographic evaluation in an attempt to provide optimal and individualized acute care. The role of serial advanced echocardiographic assessment in acute HF for risk stratification and treatment guidance has not been thoroughly explored. Methods. The “Beyond Myo-HF Study” is a prospective, non-interventional cohort trial designed to enroll acutely admitted patients with symptoms and/or signs of HF. The aim of this study is to investigate whether intrahospital changes of conventional and novel echocardiographic indices of myocardial function and congestion-related markers can predict early mortality, late mortality, and HF rehospitalization. As per the protocol, all patients undergo a pair of state-of-the-art echocardiographic assessments, with a rigorous protocol including speckle tracking analysis of all cardiac chambers and myocardial work analysis for the left and right ventricle, upon admission and pre-discharge. Their laboratory profile is captured at those two time-points, and their therapeutic management is recorded. Patients will be followed-up for a median period of 12 months after enrollment. Conclusions. The “Beyond Myo-HF” study is an ongoing, prospective trial aspiring to provide deep insight into the pathophysiology of acute HF, to enlighten the reverse cardiac functional and anatomical remodeling during hospitalization, and to recognize echocardiographic patterns capable of predicting adverse outcomes during and post decompensation of acute HF.

## 1. Introduction

Hospitalized patients with acute heart failure (HF) represent an understudied, distinct group of patients which has mainly been characterized through registries [1]. These patients largely differ from stable outpatients with chronic HF in terms of clinical course, and continue to suffer very poor prognosis; they have an estimated mortality and readmission rate of 15% and 30%, respectively, between the first and second month post discharge [2]. In this regard, it is of paramount importance to identify vulnerable subjects in order to intensify their follow-up and outpatient care. Suboptimal echocardiographic improvement in response to acute HF treatment may represent a red flag but current literature is scarce.

In acute decompensated HF, intensive therapy aims to lessen intracardiac pressure and central venous congestion [3], which is expressed as an improvement of the circulatory volume-dependent echocardiographic parameters, mainly evidenced through reverse remodeling of the thin-walled right ventricle (RV). It has been suggested that improvement of RV function surrogates such as tricuspid annular plane systolic excursion (TAPSE) and right ventricular–pulmonary artery (RV–PA) coupling is of prognostic relevance [4]. However, TAPSE has obvious limitations as it poorly reflects overall contractility of the RV [5], and it has been surpassed by speckle tracking echocardiography with longitudinal strain, which is sensitive in detecting dynamic improvement in RV mechanics [6].

Longitudinal strain does not integrate ventricular dyssynchrony into its quantitative output and does not account for the afterload, which is subject to considerable alterations with the intensive acute HF medical therapy particularly for the RV. Myocardial work is a non-invasively derived index of myocardial function that accounts both for ventricular dyssynchrony and afterload [7]. Although originally dedicated for the left ventricle, this method has recently been evaluated for the RV and has shown significant correlation with invasively measured RV stoke volume [8]. Nonetheless, its prognostic implications in the acute HF setting have not been comprehensively investigated so far.

Accordingly, using—among others—novel echocardiographic imaging biomarkers, this observational study aims to investigate (i) the intrahospital echocardiographic variations of diverse acute HF phenotypes as a response to acute treatment, (ii) the improvement or deterioration of all chambers’ systolic function parameters and echocardiographic markers of congestion during the acute HF episode, (iii) the association of admission or discharge echocardiographic parameters with early mortality, late mortality, and HF rehospitalization, and (iv) whether suboptimal echocardiographic improvement at discharge in response to acute HF treatment is associated with poor prognosis.

## 2. Methods and Materials

### Study Design and Population

The “Beyond Myo-HF Study” (ClinicalTrials.gov Identifier: NCT05573997) is an investigator-initiated, prospective, non-interventional cohort trial involving patients with acute HF. A total of 300 consecutive patients acutely admitted to the Department of Cardiology of AHEPA University General Hospital, Thessaloniki, Greece with symptoms and/or signs of HF satisfying the inclusion criteria are enrolled in the present study. Eligibility criteria are described in Table 1.

Demographic characteristics and baseline medical history including a history of previous hospitalization for acute HF, primary etiology, and clinical presentation of hospitalization are recorded for all participants. Pharmacological and non-pharmacological therapeutic maneuvers during the acute phase of hospitalization, as well as medical therapy upon admission and discharge, are collected. A comprehensive transthoracic echocardiographic assessment (TTE) is performed at two different time-points during hospitalization. The first TTE is performed within 24 h from admission, and the second with 24 h pre-discharge, to capture the trajectory and net variations of all echocardiographic indices of interest. A cardiac computed tomography or cardiac magnetic resonance study will be routinely performed as required to investigate the primary HF cause for undiagnosed patients. In addition, patients’ laboratory results including complete blood count, biochemical control, inflammatory markers, coagulation mechanism control, hormonal control, N-terminal pro-B-type natriuretic peptide, and high sensitivity troponin T levels are collected upon admission and prior to discharge (Figure 1).

Before each TTE study, heart rate and brachial blood pressure using a conventional sphygmomanometer are recorded. Additionally, height and weight of each participant are recorded and utilized to determine body mass index and body surface area in accordance with the Dubois formula [9]. Informed written consent is obtained from all participants prior to study enrollment. All patients undergo standard of care acute HF management, as dictated by their clinical presentation, the practice of the hospital, and most updated guidelines [10]. The investigation conforms with the principles outlined in the Declaration of Helsinki (2013 Amendment), and the study protocol was approved by the Ethics Committee of the School of Medicine of Aristotle University of Thessaloniki (Approval number: 19/2022, Date of approval: 10 November 2022).

In accordance with the guideline-recommended HF definitions [10], HF patients are classified on the basis of their left ventricular ejection fraction (LVEF), as this is calculated from the admission TTE study. Those are patients with (i) heart failure with reduced ejection fraction (LVEF ≤ 40%), (ii) heart failure with mildly reduced ejection fraction (LVEF = 41–49%), and (iii) heart failure with preserved ejection fraction (LVEF ≥ 50%). In addition, patient are classified as new-onset or chronic HF, on the basis of a previously documented HF hospitalization. The trend of their echocardiographic profile during hospitalization is recorded to identify predictors of early and late mortality, and HF rehospitalization at follow-up. Follow-up data are collected regularly by independent physicians via in-person or telephonic interviews. All deaths are ascertained by searching in the Greek web-based national health insurance system. Other clinical outcomes, apart from death, are identified either through hospital reports or by telephone contact.

## 3. Materials and Equipment

### 3.1. Echocardiographic Analysis

All TTE studies are performed by certified sonographers/cardiologists using high-end scanners (e.g., Vivid E95, GE Healthcare, Chicago, IL, USA). Electrocardiogram-triggered echocardiographic data are stored offline in a cine-loop format for analysis using proprietary software. All analysis is separately performed by an expert cardiologist, blinded to the clinical data of all participants.

### 3.2. Chamber Quantification Analysis

The size of all cardiac chambers is assessed in accordance with most current recommendations [11]. Two-dimensional and Doppler measurements are performed as recommended [11,12]. Simpson’s biplane method is utilized to calculate left ventricular ejection fraction (LVEF) and abnormal values of conventional left ventricular (LV) diastolic parameters are determined on the basis of recently established criteria [12]. All RV systolic function parameters including TAPSE, fractional area change, and systolic movement of the RV lateral wall (S’) are evaluated as per current guidelines [13]. RV fractional area change is calculated by the formula: Fractional Area Change = ([RV end-diastolic area − RV end-systolic area]/RV end-diastolic area) × 100%. TAPSE is measured by M-mode and S’ by pulse-wave tissue Doppler imaging (TDI) recordings of the lateral tricuspid annulus in an RV-focused view [13]. Estimated right atrial pressure is determined on the basis of the evaluation of inferior vena cava and its collapsibility [13]. Pulmonary artery systolic pressure (PASP) is derived from the tricuspid jet peak velocity and the estimated right atrial pressure using the modified Bernoulli equation [13]. An integrated approach is used to quantify the severity of valvular heart disease in line with the recommendations from the respective guidelines [14,15]. A scale ranging from 1 to 4 to define mitral regurgitation and tricuspid regurgitation severity is employed as suggested by recent published literature [15,16]: grade 1 indicates “mild”, 2 “moderate”, 3 “moderate-to-severe”, and 4 refers to “severe” mitral regurgitation or tricuspid regurgitation.

### 3.3. Strain Analysis

Two-dimensional speckle tracking echocardiography is utilized to calculate strain measurements for all four cardiac chambers. Gain settings and imaging width are adjusted to optimize the grey scale at frame rates of 50–80 Hz. On average, three different measurements are performed, using the onset of QRS as the referent point. LV, RV, and left atrial strain measurements are performed using separate proprietary software originally developed for speckle tracking analysis of each chamber. For the calculation of right atrial strain, software originally developed for the assessment of left atrial strain is utilized. For the LV, images are acquired from the apical four-chamber, apical two-chamber, and apical long-axis view. For the left atrium, images from the apical four-chamber and apical two-chamber view are employed. For the RV and right atrium, images from apical RV-focused view and apical four-chamber view, respectively, are used. Global longitudinal strain (GLS) is derived from calculation of the average of the peak systolic longitudinal strain of all segments for each chamber.

### 3.4. Left Ventricular Myocardial Work Analysis

Calculation of LV myocardial work (LVMW) is performed with the use of a commercially available vendor-specific software package. To estimate myocardial work indices, LV GLS and peak systolic LV pressure measurements are integrated with the aforementioned module, as previously described [7]. The patient’s brachial blood pressure recording is entered into the module as an estimate of peak systolic LV pressure. The latter is assumed to be equivalent to the arterial pressure, as expressed by the brachial blood pressure. Apical long-axis view is utilized to determine opening and closing timings of the aortic and mitral valves as a means to define the different phases of the cardiac cycle. Noninvasive LV pressure curves are formulated according to the duration of ejection and isovolumic phases, as determined by left-sided valve timing events. LV strain and pressure data are synchronized through alignment of the cardiac cycle phases and systolic blood pressure. Four different indices of myocardial work are calculated including (i) LV global work index (LVGWI, mmHg%), representing the total work within the LV pressure–strain loops, (ii) LV global constructive work (LVGCW, mmHg%), defined as the work performed during myocardial shortening in systole and the work during myocardial lengthening in isovolumic relaxation, (iii) LV global wasted work (LVGWW, mmHg%), representing the work contributing to the lengthening of the cardiac myocytes during systole and the shortening during isovolumic relaxation, and (iv) LV global work efficiency (LVGWE, %), defining the percentage of effectively spent work by the LV myocytes and obtained by the following formula: (LVGCW/[LVGCW + LVGWW]) × 100%.

### 3.5. Right Ventricular Myocardial Work Analysis

RV myocardial work (RVMW) indices analysis is performed by adapting proprietary software originally developed for the assessment of LVMW, as previously described [8]. Similar to the non-invasive method for evaluation of LVMW, initially developed by Russel et al. [7], the RV myocardial force–segment length loops are estimated with pressure–strain loops. PASP and pulmonary artery diastolic pressure (PADP) are entered in the software instead of systolic blood pressure and diastolic blood pressure. Pulmonary artery mean pressure (PAMP) is derived from the following formula: mean RV-right atrial gradient + mean right atrial pressure [17]. Tracing of the tricuspid regurgitation velocity–time integral is used for mean RV-right atrial pressure calculation. PADP is estimated as PADP = 1.5 × [PAMP − (PASP/3)] [13,17].

An approximation of the RV myocardial force is derived from pulmonary pressure estimates, and changes in segment length are obtained from speckle tracking echocardiography-derived strain analysis. RV global longitudinal strain (GLS), including both RV free wall and interventricular septum, are obtained from RV-focused apical four-chamber view. Pulse-wave interrogation in the basal parasternal short-axis view is applied to define timings of pulmonary valve opening and closure, whereas timings of tricuspid valve opening and closure are obtained by direct visualization of leaflets on RV-focused apical four-chamber view. RV GLS, and PASP, and PADP measurements are synchronized by right heart valve events to generate non-invasively derived pressure–strain loops for the RV. Four separate parameters of RV function are derived from the pressure–strain loops analysis. They include (i) RV global work index (RVGWI, mmHg%), calculated from the area within the global RV pressure–strain loop, (ii) RV global constructive work (RVGCW, mmHg%), an estimate of the work contributing to myocardial shortening during systole and lengthening during isovolumic relaxation, (iii) RV global wasted work (RVGWW, mmHg%), representing the work contributing to the lengthening of the cardiac myocytes during systole and the shortening during isovolumic relaxation, and (iv) RV global work efficiency (RVGWE, %), defined as the percentage of effectively spent work by the RV myocytes and obtained by the following formula: (RVGCW/[RVGCW + RVGWW]) × 100% [8].

### 3.6. Statistical Analysis

Baseline clinical and echocardiographic characteristics of each group will be examined and compared using the chi-square test for categorical variables and the two-sided Student’s *t*-test or the analysis of variance (ANOVA) test for continuous variables depending on the number of groups. Categorical variables will be represented by frequencies and percentages (%) and continuous variables will be summarized by mean ± standard deviation. Comparison of changes in echocardiographic indexes between admission and discharge will be performed using the paired *t*-test or Wilcoxon signed rank test according to the normality of distribution.

Univariate and multivariate logistic regression analysis and Cox regression analysis will be performed to identify independent predictors of outcome at follow-up. Only covariates that are significantly associated with the endpoint at univariate analysis will be included in multivariate models. The sensitivity and specificity of resulting echocardiographic parameters for the prediction of early mortality, late mortality, and rehospitalization will be evaluated with receiver operating characteristic curves. To assess the potential additive prognostic value of novel imaging biomarkers over baseline models, the additional increase of the chi-square value will be assessed using the likelihood ratio test. A time-to-event analysis will be performed according to the Kaplan–Meier method to assess whether groups of patients with specific echocardiographic patterns are associated with better or worse clinical prognosis. Event rates will be compared by the long-rank test. A two-tailed *p* value of 0.05 will be considered the significance threshold for all statistical tests. All outcomes will be reported with 95% confidence intervals. Data management and statistical analyses will be conducted using SPSS software, version 26 (IBM SPSS Statistics) and R version 3.4.4 (R Foundation for Statistical Computing, Vienna, Austria).

## 4. Discussion and Expected Results

The “Beyond Myo-HF Study” is an ongoing, prospective cohort trial of patients hospitalized for acute HF, aiming to elucidate the intrahospital echocardiographic changes and their prognostic value. By providing a comprehensive echocardiographic assessment at admission and discharge, it seeks to evaluate the acute haemodynamic changes and the subtle alterations in myocardial function by speckle tracking echocardiography, and ultimately profile subjects not responding to intensive acute HF therapy, prone to mortal events and early HF rehospitalization.

Only a few studies have described the echocardiographic changes of acute HF patients from admission to discharge [3,4,6]. Hullin et al. studied 176 AHF patients and demonstrated a significant decongestion-related improvement for mitral E wave velocity, E/e’ ratio, RV basal diameter, tricuspid regurgitation gradient, PASP, TAPSE/PASP, and inferior vena cava diameter across all LVEF groups [4]. In regard to parameters of LV systolic function, LVEF remained unchanged with decongestion while LV GLS improved, depicting an LV endomyocardial functional recovery at discharge [4]. Concerning RV function, subjects who demonstrated an increase of TAPSE and/or TAPSE/PASP with treatment had a significantly lower risk for death or cardiovascular readmission [4]. Akiyama et al. studied 77 patients with acute HF and described an improvement of E-wave deceleration time and inferior vena cava diameter for heart failure with reduced ejection fraction patients and an improvement of E/e’ for heart failure with preserved ejection fraction patients [3]. However, speckle tracking data were not provided and the prognostic implications of those changes were not evaluated [3]. Verhaert et al. studied 62 AHF patients, focusing on RV mechanics, and concluded that RV response to intensive medical therapy, indicated by improvement of RV GLS, conveyed better prognosis [6].

Both LV and RV potential functional recovery during hospitalization for acute HF is mainly detected by longitudinal strain of LV and RV, respectively, by applying speckle tracking echocardiography [4,6]. This parameter is more capable of tracking subtle intrinsic myocardial response to treatment compared to LVEF [18], and in an angle-independent manner compared to Doppler-derived indices of ventricular function. Although longitudinal strain evaluates the endomyocardial function and is not merely based on volumetric ventricular changes like RV fractional area change or LVEF [19], it is not completely independent of afterload. However, in acute HF the afterload of both LV and RV may change significantly during treatment with inotropes and heavy dosing of diuretics, while a reduction of blood pressure and pulmonary pressure is driven mainly by changes to the patient’s volume status and vascular resistance.

Especially for those on inotropes in the acute HF setting, suboptimal echocardiographic improvement of the intrinsic myocardial systolic properties to therapy may indicate limited myocardial reserve for the existent afterload [6]. In patients with coexistent significant mitral or tricuspid regurgitation, the longitudinal strain may be more sensitive for revealing myocardial functional improvement compared to volumetric assessment of the ventricular function [20], considering that the ventricle partly empties in a low-pressure cavity; of note, the grade of the regurgitation—especially if it is functional—is substantially dependent on the afterload. Thus, there is an unmet need to adjust the longitudinal strain for the ventricular afterload in acute HF patients, which is currently feasible by echocardiography-evaluated myocardial work indices [7].

In the current study, the response to acute HF therapy will be assessed, among others, by computation of non-invasive LV myocardial work parameters. Such novel indices are estimated via dedicated software by integrating speckle tracking LV GLS, blood pressure as an afterload estimate, and cardiac event timing to formulate a pressure–strain loop of the LV [7]. Accounting for dyssynchrony and afterload, LV global myocardial work has shown to provide incremental prognostic information over and above LV GLS and LVEF regarding all-cause mortality or HF hospitalization between subjects with heart failure with reduced ejection fraction [21]. Additionally, Hedwig et al. displayed that LV GWI and LV GCW could independently predict events among a group of advanced HF patients [22]. Only one study has assessed dynamic in-hospital changes of myocardial work indices for acute HF patients and observed improvement of GCW and GWI only for heart failure with reduced ejection fraction but not heart failure with preserved ejection fraction patients [23]. The prognostic implications of those intrahospital changes were not investigated in the aforementioned work [23].

The RV is a thin-walled cardiac chamber with low ventricular elastance, whose contractile properties are more susceptible to its loading environment, preload, and afterload, compared with the LV [24]. Those characteristics render the RV more vulnerable to acute changes in hemodynamic conditions and thus suited for serial echocardiographic assessment to identify response to intensive therapy and recovery after an acute HF episode [6]. Reducing peripheral congestion or pulmonary pressures would have a strong impact on RV size and function. RVMW, combining information on contractile function by stain analysis and on afterload assessed by PASP, is an echocardiographic parameter that could reflect alterations in response to acute treatment. This study aims to expand the current literature by investigating the intrahospital changes and prognostic significance of RV myocardial work in acute HF.

Functional tricuspid regurgitation is a very common valvular heart disease growing in parallel with the progression of HF [25]. Significant functional TR is associated with unfavorable natural history for patients with HF, as it leads to a vicious cycle of RV dilatation, arrhythmias, and RV failure when left untreated [26]. However, functional tricuspid regurgitation is a marker of congestion and its dynamic nature with improvement in response to medical therapy should be acknowledged. This improvement is theoretically driven by adverse RV remodeling with reduction of the RV afterload, the RV and tricuspid annular dimensions, attenuation of the tethering effects, and functional RV recovery. Among others, this study aims to identify the RV anatomical and functional profile of subjects who experience amelioration of tricuspid regurgitation grade in response to acute HF therapy, and explore if this profile is associated with better outcomes.

## 5. Limitations

Some limitations of the study should be acknowledged. Although the study is prospective, and includes a state-of-the-art, rigorous echocardiographic protocol twice during hospitalization, its single-center character necessitates the confirmation of the results from other centers to ensure their clinical validity. Additionally, the association between intrahospital echocardiographic changes and therapeutic interventions is challenging, considering that therapeutic decisions are personalized and may change on a daily basis during acute HF treatment.

## 6. Conclusions

In conclusion, acute HF management remains an understudied subject in need of further attention. Through serial assessment of the echocardiographic profile and the introduction of novel imaging biomarkers, this study has the potential to enlighten the pathophysiology of acute HF, provide guidance to individualized management, and ultimately identify early vulnerable patients requiring escalation of therapeutic decisions and close follow-up.

## Figures and Tables

**Figure 1 diagnostics-13-01191-f001:**
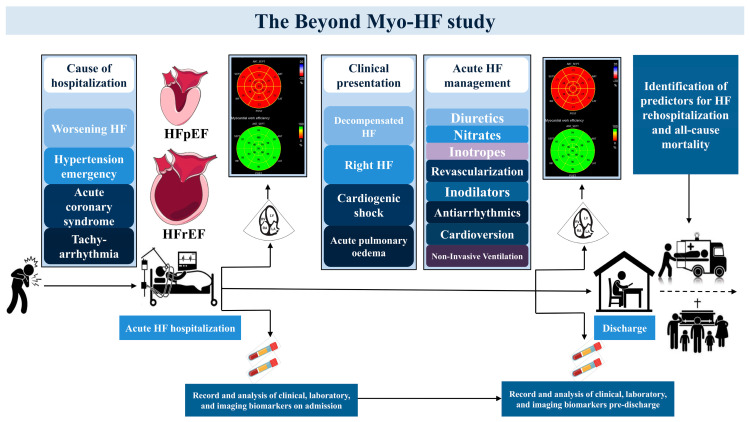
Study design diagram. Serial echocardiographic and laboratory assessment will be performed in 300 consecutive patients admitted with acute heart failure, in order to identify predictors of early mortality, late mortality, and heart failure rehospitalization. Abbreviations: ACS, acute coronary syndrome; HFpEF, heart failure with preserved ejection fraction; HFrEF, heart failure with reduced ejection fraction.

**Table 1 diagnostics-13-01191-t001:** Inclusion and exclusion criteria of the “Beyond Myo-HF study”.

Inclusion Criteria	Exclusion Criteria
Symptoms and/or signs of heart failure Abnormal plasma concentration of N-terminal pro-B-type natriuretic peptide, measured within 24 h of admission Objective echocardiographic evidence of cardiac structural and/or functional abnormalities consistent with the presence of LV systolic dysfunction/diastolic dysfunction/raised LV filling pressures upon admission assessment Adult patients	Symptoms and/or signs of HF secondary to congenital heart disease, infective endocarditis, pericardial disease, and history of recent cardiac surgery (<1 month); pulmonary circulation pathologies such as I, IV, V pulmonary hypertension Patients receiving maintenance hemodialysis or peritoneal dialysis Severe liver insufficiency Patients with active malignancy and/or life expectancy <1 year because of comorbidities Heart transplantation Poor echocardigraphic acoustic windows precluding reliable assessment and/or analysis Patients not providing informed written consent to participate in the study

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
