# Peer review of "Clinical Value of Novel Echocardiographic Biomarkers Assessing Myocardial Work in Acute Heart Failure—Rationale and Design of the “Beyond Myo-HF Study”"

_diagnostics, 2023, doi:10.3390/diagnostics13061191_

Round 1
Reviewer 1 Report (Previous Reviewer 1)
We would note that the required revisions were made in the manuscript so that it is clear that it considers echocardiographic changes that can serve as biomarkers for acute heart failure. Having said that, the article can be published in its current form.
Reviewer 2 Report (Previous Reviewer 2)
Thank you for revised and response. The manuscript was more valuable.
Reviewer 3 Report (New Reviewer)
The authors in this paper report the design of the Beyond Myo-HF study. The authors aim to assess whether echocardiographic patterns are capable of predicting adverse outcomes during an admission for decompensated heart failure. Although multiple studies have assessed echocardiographic variables and impact on HF, this study aims to do it in a prospective manner with assessment of prognostic variables. Will be interesting to see the results of this study.
Though not discussed, the impact of echo on therapeutic decisions ie if echocardiographic changes management will also add insight to study and whether repeating echo during a hospital stay will impact change in management can add insight in addition to the prognostic variables.
Minor edits
-Title - ‘’Beyond Myo-HF study’’ ; quotes starting are inverted.
This manuscript is a resubmission of an earlier submission. The following is a list of the peer review reports and author responses from that submission.
Round 1
Reviewer 1 Report
The study carried out by your team is interesting and well documented, showing advanced concerns for the study of echocardiographic changes in patients with heart failure.
I noticed that your study includes patients with acute heart failure. Here, you should have emphasized whether this acute heart failure is the first manifestation of a heart condition or is an exacerbation of a chronic heart failure.
In addition, modern trends are to perform more advanced imaging explorations such as cardiac CT or cardiac MRI.
Perhaps these explorations cannot be performed in a patient with acute heart failure but can be performed later. Thus, the clinical value of the echocardiographic examination in the episode of acute heart failure would increase in accuracy if it benefited from a subsequent monitoring by CT or cardiac MRI.
Since you mention in the study that it was carried out on patients with acute heart failure, I believe that this detail should be specified in the title of the article.
Reviewer 2 Report
The manuscript entitled "Clinical Value of Novel Echocardiographic Bi-omarkers assessing the Myocardial Work in Heart Failure – Rational and design of the ‘’Beyond Myo-HF study’’ aims to investigate whether intra-hospital changes of conventional and novel echocardiographic indices of myocardial function, and congestion-related markers can predict early, late mortality, and HF rehospitalization. The topic is interesting and the manuscript is well written, however, resubmitt after result available is more suitable.